# The First Two Cases of Monkeypox Infection in MSM in Bahia, Brazil, and Viral Sequencing

**DOI:** 10.3390/v14091841

**Published:** 2022-08-23

**Authors:** Carlos Brites, Felice Deminco, Marcia Sampaio Sá, Jean Tadeu Brito, Estela Luz, Andreas Stocker

**Affiliations:** LAPI—Laboratório de Pesquisa em Infectologia, Hospital Universitário Professor Edgard Santos, Faculdade de Medicina da Bahia, Salvador 40110060, Brazil

**Keywords:** monkeypox, HIV, MSM, Brazil

## Abstract

Monkeypox infection is rapidly spreading across the world. Despite the increasing number of cases, only a few reports have been published, and most are on people living without HIV. We report here the first two cases of monkeypox infection in Bahia, Brazil, one of them in a person living with HIV, on stable treatment. Both cases had a similar evolution, with a limited number of lesions and mild symptoms, with a complete recovery after 7–10 days. The potential route of transmission was via oral sex for the first case and was undefined for the second one. Both cases were confirmed through detection of the viral genome by PCR, and the partial sequence of the first case indicates the infection was caused by the West African clade. These cases confirm that monkeypox infection is currently being transmitted in Brazil and that people living with HIV on stable treatment are not likely to present a more severe form of monkeypox.

## 1. Introduction

### Monkeypox

Monkeypox is an uncommon viral disease, caused by an *Orthopoxvirus*, family Poxviridae. It is an enveloped, double-stranded DNA virus that belongs to the same genus that includes variola, vaccinia and cowpox viruses. Although the source of the disease remains unclear, the virus can infect African rodents and non-human primates (such as monkeys) as well as humans. Monkeypox was first described in 1958, after two outbreaks in colonies of monkeys used for scientific studies, in Copenhagen, Denmark [1].

The monkeypox virus has two distinct genetic clades, the Central African (Congo Basin) clade and the West African clade. The first one (Congo Basin) is linked to more severe disease and seems to be more transmissible [2]. The only country where the two clades coexist is Cameroon.

## 2. Human Infection

The first monkeypox case in humans was reported in 1970. Since then, many outbreaks were reported, mostly in African countries, with occasional reports in other regions [3,4,5]. The last reported outbreak of human infection outside Africa occurred in 2003 [6]. Transmission of the monkeypox virus from one person to another is associated with close contact with lesions, respiratory droplets, body fluids and contaminated materials such as bedding. The usual average incubation period of monkeypox varies from 6 to 13 days but can range from 5 to 21 days.

Monkeypox is usually a self-limited disease. Most cases present with mild symptoms, and a there is a complete recovery in about two weeks. However, children, pregnant women, or people with immune suppression due to other health conditions can present with a more severe disease, especially those infected with the Congo Basin clade. The fatality rate increases from 3.6% (for the West African clade) to 10.6% in cases of infection by the Congo Basin clade [2].

### 2.1. Case Report 1

A 37-year-old MSM living with HIV since 2012 on stable cART (lamivudine + tenofovir + efavirenz in a fixed-dose combination pill) and undetectable HIV-1 RNA plasma viral load since the beginning of treatment came to a medical visit reporting that 3 days ago, he developed night fever (38.5 °C), chills and generalized muscle pain. His last CD4-positive cell count was 604 (41%), in December 2021. He also noted a urethral burning sensation during urination. The following morning, he noted small papules/vesicles, on an erythematous base, on the glans and scrotal sac. Five other small lesions were observed on his forehead, nose, thorax, and left leg. The lesions were mildly painful, without pruritus. He was initially attended by a urologist who requested tests for sexually transmitted infection (molecular detection tests for Chlamydia, *Mycoplasma*, genital herpes virus and HPV) and prescribed doxycycline (100 mg bid, for 10 days) plus ceftriaxone (1 g IM, single dose). He started treatment and scheduled a next day visit to an infectious disease specialist. He has been engaged in a stable relationship with the same partner for over 15 years. He had a business trip to Europe (Italy) in the last 30 days, returning to Brazil 22 days before the onset of symptoms. He denied any close contact with other people during the traveling period. However, he had received oral sex from an occasional partner 14 days before symptom onset and had no information on the current health conditions of that partner. He only knew that the occasional partner was a frequent traveler, with frequent trips to other Brazilian regions and other countries.

He did not use psychoactive substances and reported only moderate alcohol intake. He never smoked. During the medical evaluation he was in good clinical condition, had normal temperature and presented with the lesions described above (Figure 1 and Figure 2); some of them were ulcerated but showed no signs of bacterial infection or crusts. The only additional finding at physical examination was lymphadenopathy, with enlargement and tenderness of inguinal lymph nodes, the largest measuring 1.2 × 0.5 cm. We suspected monkeypox infection and asked for a molecular detection test. The PCR was carried out at the Laboratório de Pesquisa em Infectologia (LAPI), at Hospital Universitário Professor Edgard Santos, Federal University of Bahia, as described in the next section. The lesions began to heal in the following days, and after one week of symptoms onset, he showed only a residual process, as seen in Figure 2.

### 2.2. Case Report 2

The patient was a 31-year-old MSM reporting sudden onset of fever, headache, back pain and lymph node enlargement in the cervical region, followed by a cutaneous, papular–vesicular rash that started 4 days before the medical visit. During the medical evaluation, he was in good clinical condition and had normal temperature. The lesions were predominantly on his head, but isolated lesions were detected on the legs, trunk, hands, and perianal area. He did not use psychoactive substances and never smoked, reporting only moderate alcohol intake. He reported no recent trip or sexual contact with new partners in the last 3 weeks. He is engaged in a stable relationship with another man and declared no close contact with other people in the last 2 weeks. He persisted with active lesions for one additional week, without any other clinical event. Figure 3 shows some of the lesions detected during physical examination. We suspected monkeypox infection and asked for a molecular detection test. A swab collection of vesicles’ contents and crusts tested positive for monkeypox. The PCR was carried out at the Laboratório de Pesquisa em Infectologia (LAPI), at Hospital Universitário Professor Edgard Santos, Federal University of Bahia. The value of the threshold cycle for this case was 18.29.

Both patients provided written informed consent. The case reports were approved by the institutional committee on ethics in research.

## 3. Laboratorial Diagnostic Procedures

Two sample swabs of the skin lesions’ secretions were collected and kept in viral transport solution. After 15 min of incubation at room temperature, the material was shortly vortexed and aliquoted. The original tube with swabs and 200 µL were immediately extracted and purified by PureLink Genomic DNA Mini Kit/Invitrogen. Some aliquots were kept in an ultra-freezer at −80 °C for further analyses in the reference lab (Fiocruz/Rio de Janeiro, RJ, Brazil). The analysis was performed using two modified TaqMan qPCR assays, already published [7]. One assay is specific for the Western African (WA) MPXV, and the other is a generic qPCR for all MPXV. To reduce the potential risk of mispriming caused by mutations, we did not use the TaqMan probes and used an unspecific SYBR Master Mix (Power SYBR™ Green PCR Master Mix Applied Biosystems™, Waltham, MA, USA). The run was performed on a 7500 Real-Time PCR System (Applied Biosystems, Waltham, MA, USA). Mixes and temperature cycles were adapted, and the PCR reaction of either 25 µL of WA or generic assay contents, 12.5 µL of Power SYBR Green PCR Master Mix (2×), 1.0 µL of BSA 1 mg/mL (0.04 µg/µL), 0.75 µL of forward and reverse primer 10 µM solution (0.3 µM), 5 µL of ultrapure PCR water to complete the volume and 5 µL of extracted sample DNA was carried out. The run was performed with an initial denaturation of 95 °C for 10 min and 45 two-step cycles with 20 s of 94 °C and 40 s of 56 °C followed by a melting curve.

To evaluate the results by Sanger sequencing, the amplicons were purified using a PureLink PCR Purification Mini Kit/Invitrogen, quantified by IQuant Touch/Loccus (WA: 19 ng/µL; generic: 27 ng/µL) and labeled in a 10 µL BigDye Terminator 3.1/Applied Biosystems reaction using 1 µL of Big Dye Terminator solution (2.5×), 1.5 µL reaction buffer (5×), 2.0 µL of forward or reverse primer 2.0 µM (0.4 µM), 3.5 µL of ultrapure PCR water to complete the volume and 2.0 µL of purified PCR product. The run was performed in a Mastercycler Gradient/Eppendorf using denaturation of 96 °C for 1 min and 25 three-step cycles of 96 °C for 10 s, 50 °C for 5 s and 60 °C for 4 min. After purification by isopropanol 75% precipitation, sequencing was performed in a SeqStudio/Applied Biosystems using the short-sequence run protocol. The quality of chromatograms using Basecaller/Applied Biosystem values was evaluated using TREV Software (Staden Package), and sequences were aligned with four reference sequences from the original publication and evaluated in BioEdit software [7].

The value of the threshold cycle for the first case was 16.06 for both assays. In addition, the melting curve was clean and showed, for every assay, a single product; Tm = 75.0 °C for the WA assay and Tm = 75.7 °C for the generic assay (Figure 4).

It could be shown that the two PCR products were a 100% match with the published MPXV sequences (Figure 4 and Figure 5) [7].

## 4. Discussion and Literature Review

The first human case of monkeypox was recorded in 1970 [3]. Since then, monkeypox has been reported in people in central and western African countries as well as in the USA [4,5,6]. Prior to the 2022 outbreak, nearly all monkeypox cases in people outside of Africa were linked to international travel to countries where the disease commonly occurs or to imported animals.

Since 13 May, an increasing number of human monkeypox cases have been reported to the WHO, and the cases have rapidly spread to different countries and regions. Most of cases have been associated with traveling to another country and were also predominantly detected in MSM. In addition, the lesions were detected mainly in the perineal/genital areas in association with inguinal lymphadenopathy [8].

Only a few reports on monkeypox in people living with HIV (PLWH) have been published. To date, there is little available information on the disease course in such a population—only three cases reported on patients in Romania, Czech Republic and Portugal [9,10,11]. One of these reports describes monkeypox coinfection with syphilis while another describes a case of monkeypox and acute HIV coinfection. In Brazil, over 800 cases of monkeypox have been reported to public health authorities so far, most of them among MSM, but we were not able to find information on the infection among PLWH. We did not find previous reports on monkeypox cases in Brazil, and there are only a few published cases on monkeypox infection in PLWH worldwide.

In the present report, we describe a monkeypox case in a PLWH and a second case in a HIV-negative subject. Both cases showed a mild presentation of disease and prompt recovery, suggesting that the monkeypox infection’s natural course probably is not changed in stable, on treatment, PLWH. The clinical presentation was quite similar for both patients, reinforcing the idea that controlled HIV infection is not a risk for a more severe monkeypox disease. In addition, the history of case 1 suggests that receptive oral sex can be a potential route of transmission, because it was the only exposure risk reported by the patient. The fact that his steady partner was apparently not infected by monkeypox reinforces the available evidence informing that asymptomatic patients are less able to transmit the infection, even to close contacts. The second case also indicates that community transmission is already occurring in Brazil, because the patient had no history of recent travel or sexual contact with occasional partners. In addition, his sexual partner was asymptomatic, but he was not tested for monkeypox. Both cases occurred in MSM, but the absence of a clear sexual exposure in the second case suggests that there may be other routes of transmission, as already indicated by some authors [12,13,14,15].

The genetic sequencing of the virus confirms that it belongs to the Western African clade, the less aggressive viral clade. There is still no available information on the outcomes of Congo Basin clade infections in PLWH. Monkeypox has already been declared as a world emergence by the WHO, and the fast spread of cases in Brazil is a concerning problem for health authorities. To date, there is no established treatment for monkeypox, and vaccines against it are not widely available. Preventive strategies must be implemented to allow the proper control of the current outbreak and to identify potential epidemiological changes that increase the risk of monkeypox dissemination to non-endemic countries and make its control difficult [15]. In addition, the clinical evolution and the outcome of monkeypox among PLWH without treatment or with uncontrolled disease deserve further studies.

## Figures and Tables

**Figure 1 viruses-14-01841-f001:**
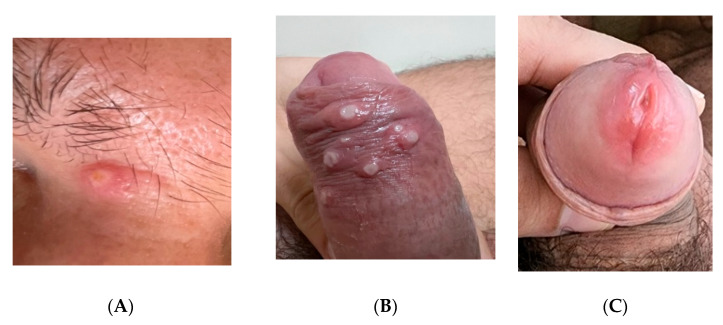
Acute lesions presented by the first case (**A**–**C**).

**Figure 2 viruses-14-01841-f002:**
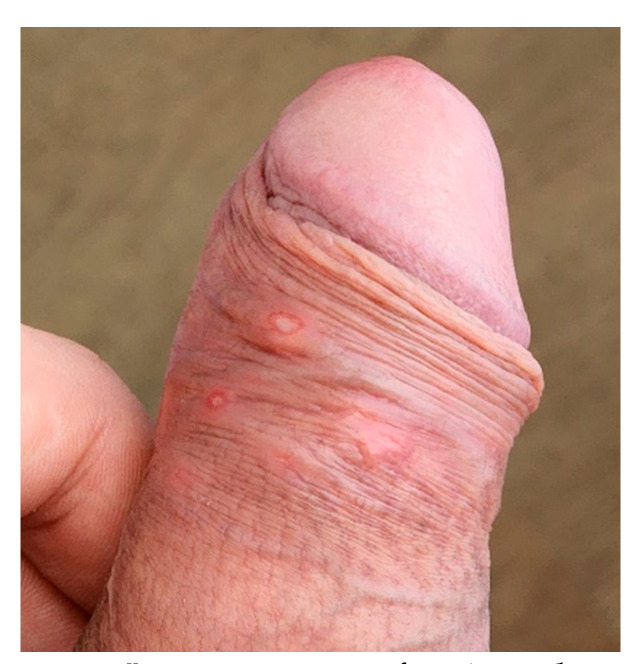
Lesions aspect after 1 week.

**Figure 3 viruses-14-01841-f003:**
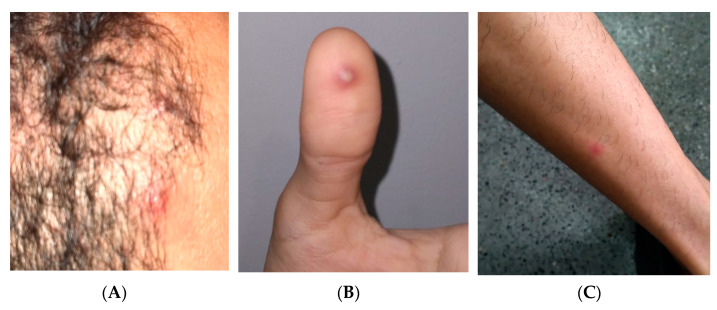
Lesions presented by case 2 at medical visit (**A**–**C**).

**Figure 4 viruses-14-01841-f004:**
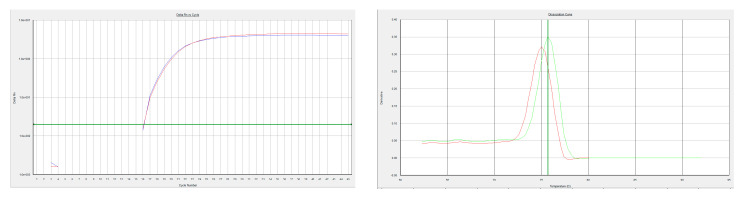
Amplification and melting curves of WA. Amplification curves of WA (red, Ct = 16.06) and generic (blue, Ct = 16.06) assay. Melting curve of WA (Tm = 75.0 °C) and generic (Tm = 75.7 °C) assay.

**Figure 5 viruses-14-01841-f005:**
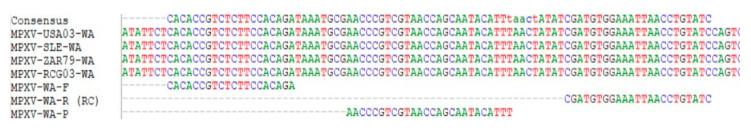
Previously published MPXV sequence. Alignment of the sequenced WA assay product (consensus) with reference sequences and PCR oligos. Alignment of the sequenced generic assay product (consensus) with reference sequences and PCR oligos.

## Data Availability

All data are available under request to the corresponding author.

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
