# Peer review of "The First Two Cases of Monkeypox Infection in MSM in Bahia, Brazil, and Viral Sequencing"

_viruses, 2022, doi:10.3390/v14091841_

Round 1
Reviewer 1 Report
In this paper Carlos Brites et al reported 2 MPX cases one in an HIV infected patient and the other in a non-HIV patient. It is well-written and documented. I have few comments and suggestions:
1. I think that is no need to report if the patients have been vaccinated against SARS-CoV-2, maybe if they are vaccinated against smallpox or chickenpox.
2. About the second case, authors reported that the sexual partner was asymptomatic, this means that was he tested for MPX or not? Please specify this in your paper.
3. I did not find in references chapter the reference about romanian case (Oprea C, Ianache I, Piscu S, Tardei G, Nica M, Ceausu E, Popescu CP, Florescu SA. First report of monkeypox in a patient living with HIV from Romania. Travel Med Infect Dis. 2022 Jun 24;49:102395. doi: 10.1016/j.tmaid.2022.102395. Epub ahead of print. PMID: 35753658.)
Author Response
Reviewer 1
We thank reviewer 1 for the useful comments and sugestions. Please see below a point-by-point response to the questions.
In this paper Carlos Brites et al reported 2 MPX cases one in an HIV infected patient and the other in a non-HIV patient. It is well-written and documented. I have few comments and suggestions:
- I think that is no need to report if the patients have been vaccinated against SARS-CoV-2, maybe if they are vaccinated against smallpox or chickenpox.
R- We deleted the information on SARS-CoV-2 vaccine. They were not vaccinated against smallpox or checkenpox
- About the second case, authors reported that the sexual partner was asymptomatic, this means that was he tested for MPX or not? Please specify this in your paper.
R- The partner was not tested, because no lesion was detected, and he was completely asymptomatic. We added the information on testing to text.
- I did not find in references chapter the reference about romanian case (Oprea C, Ianache I, Piscu S, Tardei G, Nica M, Ceausu E, Popescu CP, Florescu SA. First report of monkeypox in a patient living with HIV from Romania. Travel Med Infect Dis. 2022 Jun 24;49:102395. doi: 10.1016/j.tmaid.2022.102395. Epub ahead of print. PMID: 35753658.)
R- Thanks for the note, the reference was added to reference list
Reviewer 2 Report
Dear Authors ,
Thank you for your efforts in presenting the case report. Here are my comments to be considered in your revision :
- The manuscript title must be improved.
- The paper can benefit from these articles in the background and/or discussion:
https://doi.org/10.3389/fitd.2022.951380
https://doi.org/10.3390/medicina58070924
- Information about ethical approval / patients' consent have to be presented
- I recommend the paper to go through English language editing.
Best regards
Author Response
We thank reviewer 2 for the comments and suggestions that surely will help to improve the final manuscript
Reviewer 2
Thank you for your efforts in presenting the case report. Here are my comments to be considered in your revision :
- The manuscript title must be improved.
R- We modified the title to make it more precise
- The paper can benefit from these articles in the background and/or discussion:
https://doi.org/10.3389/fitd.2022.951380
https://doi.org/10.3390/medicina58070924
Thanks for the suggestions. We include the first reference, as it is an interesting review on monkeypox, but the second one does not fit well in our cases report.
- Information about ethical approval / patients' consent have to be presented
R- ethical approval was added to the text
- I recommend the paper to go through English language editing.
R- English language was reviewed as suggested